# Mechanisms Underlining Inflammatory Pain Sensitivity in Mice Selected for High and Low Stress-Induced Analgesia—The Role of Endocannabinoids and Microglia

**DOI:** 10.3390/ijms231911686

**Published:** 2022-10-02

**Authors:** Piotr Poznanski, Joanna Giebultowicz, Justyna Durdzinska, Tomasz Kocki, Mariusz Sacharczuk, Magdalena Bujalska-Zadrozny, Anna Lesniak

**Affiliations:** 1Department of Experimental Genomics, Institute of Genetics and Animal Biotechnology, Polish Academy of Sciences in Jastrzebiec, Postepu 36A Str., 05-552 Magdalenka, Poland; 2Department of Bioanalysis and Drugs Analysis, Faculty of Pharmacy, Medical University of Warsaw, Centre for Preclinical Research and Technology, Banacha 1 Str., 02-097 Warsaw, Poland; 3Department of Pharmacodynamics, Faculty of Pharmacy, Medical University of Warsaw, Banacha 1 Str., 02-097 Warsaw, Poland; 4Department of Experimental and Clinical Pharmacology, Medical University of Lublin, Jaczewskiego 8b Str., 20-090 Lublin, Poland

**Keywords:** anandamide, 2-arachidonoylglycerol, HA/LA mice, microglia, receptor expression

## Abstract

In this work we strived to determine whether endocannabinoid system activity could account for the differences in acute inflammatory pain sensitivity in mouse lines selected for high (HA) and low (LA) swim-stress-induced analgesia (SSIA). Mice received intraplantar injections of 5% formalin and the intensity of nocifensive behaviours was scored. To assess the contribution of the endocannabinoid system, mice were intraperitoneally (i.p.) injected with rimonabant (0.3–3 mg/kg) prior to formalin. Minocycline (45 and 100 mg/kg, i.p.) was administered to investigate microglial activation. The possible involvement of the endogenous opioid system was investigated with naloxone (1 mg/kg, i.p.). Cannabinoid receptor types 1 and 2 (*Cnr1*, *Cnr2*) and opioid receptor subtype (*Oprm1*, *Oprd1*, *Oprk1*) mRNA levels were quantified by qPCR in the structures of the central nociceptive circuit. Levels of anandamide (AEA) and 2-arachidonoylglycerol (2-AG) were measured by liquid chromatography coupled with the mass spectrometry method (LC-MS/MS). In the interphase, higher pain thresholds in the HA mice correlated with increased spinal anandamide and 2-AG release and higher *Cnr1* transcription. Downregulation of *Oprd1* and *Oprm1* mRNA was noted in HA and LA mice, respectively, however no differences in naloxone sensitivity were observed in either line. As opposed to the LA mice, inflammatory pain sensitivity in the HA mice in the tonic phase was attributed to enhanced microglial activation, as evidenced by enhanced *Aif1* and *Il-1β* mRNA levels. To conclude, Cnr1 inhibitory signaling is one mechanism responsible for decreased pain sensitivity in HA mice in the interphase, while increased microglial activation corresponds to decreased pain thresholds in the tonic inflammatory phase.

## 1. Introduction

Numerous reports dedicated to studying the role of the endocannabinoid system in pain modulation have unequivocally determined that the enhancement of endocannabinoid tone produces an anti-inflammatory phenotype in animal models or pain insensitivity in human subjects [1,2,3,4,5]. Endocannabinoid modulation of the inflammatory response includes cannabinoid receptor 2 (CNR2)-mediated suppression of proinflammatory cytokines (e.g., IL-1β, IL-6, IL-8), suppression of dendritic cell activation and mast cell maturation as well as cannabinoid receptor 1 (CNR1)-dependent actions like mobilization of myeloid-derived suppressor cells (MDSCs) or inhibition of the contractile response [6]. Recent findings have demonstrated that agonists with dual CNR1 and CNR2 activities show promise with regard to quenching the neuroinflammatory and excitotoxic cascades in multiple sclerosis [7]. Moreover, the new trend in cannabinoid drug design also features bitopic molecules that target both the orthostatic and allosteric CNR2 sites for enhanced anti-inflammatory activity and lower side effect liability [8]. Positive allosteric modulation of the CNR1 receptor was also proven effective in alleviating inflammatory pain [9].

Blocking endocannabinoid activity by directly targeting both subtypes of cannabinoid receptors (CNR1 and CNR2) with antagonists/inverse agonists or by hindering their breakdown via the inhibition of fatty acid amide hydrolase (FAAH) or monoacylglycerol lipase (MAGL) are common approaches for studying the importance of the endocannabinoid system in pain sensitivity. Genetically engineered cell-type specific knock-in or knock-down models or models with disrupted FAAH, MAGL or CNR receptor subtype function are also popular [10,11]. However, some reports point to the disadvantages of such pharmacologic or genetic manipulations, which include no sustained analgesic activity or paradoxical exacerbation of inflammatory or neuropathic pain [12,13,14].

Rodent models showing divergent endocannabinoid system activity as a result of stringent, multigenerational selection protocols show a more stable phenotype, thus the employment of such models opens new possibilities for unraveling the complex mechanisms that govern pain sensitivity. Two mouse lines selected for over 100 generations to express either depressed (HA—high analgesia line) or enhanced (LA—low analgesia line) pain sensitivity show divergent activity in the two important neurotransmitter systems engaged in pain perception. Namely, the HA line shows a greater contribution of the opioid and cannabinoid systems in the acute antinociceptive response to noxious or stressful stimuli and are also more sensitive to exogenous opioid and cannabinoids ligands [15,16,17,18].

Additionally, HA mice were also described to show a more profound inflammatory pain phenotype in comparison with their LA counterparts, possibly owing to the previously discovered congenital blood–brain barrier (BBB) disruption in this line [19,20]. Thus, the notion that HA mice might show enhanced microglial activation, particularly in response to inflammatory stimuli, seems plausible. Endocannabinoids are vital players in promoting microglial activation directed towards tissue remodeling to maintain central nervous system (CNS) homeostasis [21,22]. Similar observations were made for dynorphins and enkephalins acting via κ and δ opioid receptors, respectively [23,24]. Therefore, one can expect that upregulation of the endocannabinoid and endogenous opioid system in HA mice could represent a compensatory mechanism that counteracts the pro-inflammatory microglial activation.

There were previous attempts to confirm that the mechanistic underpinnings of increased susceptibility to inflammatory pain is linked to endogenous opioid system activity. It is a widely accepted belief that endogenous opioids exert anti-inflammatory activity mainly by suppressing the stress response pro-inflammatory cytokine positive feedback loop [25]. Endogenous opioids can also inhibit splenocyte proliferation and T-cell activation [26,27]. As shown before, downregulation of enkephalin expression correlates with the upregulation of pro-inflammatory markers in a number of inflammatory diseases like arthritis, Crohn’s disease or dry eye disease [28,29,30]. Additionally, the phenomenon of downregulating IL-1 family cytokines is thought to constitute the basis for placebo-based analgesia, where an increase in endogenous opioid tone is key.

Surprisingly, the results obtained so far from HA and LA mice did not confirm any meaningful engagement of endogenous opioid peptides in inflammatory pain sensitivity as previously hypothesized [31]. Moreover, no attempts have been made so far to explain the role of endogenous opioids and cannabinoids in transient dorsal horn neuron hyperpolarization following irritant injection. Thus, in the study herein presented, we aimed at verifying the hypothesis that differences in tonic inflammatory pain sensitivity and the magnitude of transient neuron irresponsiveness after formalin injection between HA and LA mice emerged from the divergent endocannabinoid system tone and possibly from differences in congenital BBB function. We believe that increased endocannabinoid, rather than endogenous opioid system tone, serves as a mechanism compensating for microglial-activation-induced increased inflammatory pain sensitivity in HA mice. Additionally, we showed that endocannabinoids suppressed the transient inhibition of neuron excitability in response to formalin injection.

## 2. Results

### 2.1. Sensitivity to Acute Inflammatory Pain in HA and LA Mice

As revealed by two-way ANOVA with line and phase as the independent factors, mice from the HA and LA lines differed in the intensity of nocifensive responses (measured as paw licking time) to formalin injection in the interphase and second phase of the formalin test [F(1,41) = 43.2; *p* ≤ 0.001] (Figure 1). A significant line x phase interaction [F(1,40) = 72.2; *p* ≤ 0.001] revealed differences in the intensity of the nocifensive reaction to formalin injection between the lines. The HA mice spent significantly more time licking the injured paw in the second phase (267 ± 20 s vs. 167 ± 12; *p* ≤ 0.001) than did their LA counterparts [F(1,20) = 21.0; *p* ≤ 0.001]. Surprisingly, during the interphase, the HA mice spent significantly less time licking the injured paw (16 ± 4 s vs. 121 ± 8 s; *p* ≤ 0.001) [F(1,20) = 104.2; *p* ≤ 0.001].

### 2.2. Role of the Endogenous Opioid System in Acute Inflammatory Pain Sensitivity in HA and LA Mice

Two-way ANOVA with treatment and phase as the independent factors revealed that the administration of naloxone (NLX) did not affect licking time in the HA [F(1,37) = 0.09; *p* = 0.92] nor in the LA mice [F(1,29) = 0.21; *p* = 0.64] (Figure 2A,B). Only a tendency for increased paw licking was noticed in the NLX-treated HA mice when compared to mice receiving saline, but it failed to reach statistical significance.

### 2.3. Role of the Endocannabinoid System in Acute Inflammatory Pain Sensitivity in HA and LA Mice

Two-way ANOVA with treatment and phase as the independent factors revealed that when the HA mice were challenged with a single dose of a selective Cnr1 receptor antagonist, rimonabant (0.3–3 mg/kg, RIM), there was a significant change in licking time [F(3,67) = 8.4; *p* ≤ 0.001] (Figure 3A). Additionally, the effect of rimonabant was dependent on the phase of the test [F(1,67) = 88.6; *p* ≤ 0.001]. An increase in licking time was observed in the interphase in the HA mice receiving 1 mg/kg (163 ± 20 s vs. 16 ± 4 s; *p* ≤ 0.001) and 3 mg/kg of rimonabant (158 ± 22 s vs. 16 ± 4 s; *p* ≤ 0.01) as compared with the controls [F(3,32) = 28.6; *p* ≤ 0.001]. On the other hand, rimonabant did not affect licking time in the HA mice in the second phase of the formalin test [F(3,32) = 1.6; *p* = 0.20].

In the LA line, rimonabant treatment also resulted in an increase in licking time and its effect was phase-dependent, as reported by two-way ANOVA [F(3,72) = 6.6; *p* ≤ 0.001] (Figure 3B). As opposed to the HA mice, paw licking was unaffected by rimonabant in the LA mice in the interphase, as indicated by one-way ANOVA [F(3,35) = 1.86; *p* = 0.15]. Again, as opposed to the HA mice, all three doses of rimonabant increased paw licking time in the LA mice in the second phase of the test from 167 ± 12 s to 261 ± 28 s (*p* ≤ 0.001), 236 ± 20 s (*p* ≤ 0.01) and 293 ± 23 s (*p* ≤ 0.001), respectively [F(3,35) = 1.86; *p* ≤ 0.001]. Phase-dependent alterations in nocifensive behaviour following rimonabant treatment between lines were confirmed by a positive treatment x phase x line interaction [F(3,133) = 3.9; *p* ≤ 0.01].

### 2.4. Cannabinoid Cnr1 and Cnr2 Receptor Expression in HA and LA Mice

Two-way ANOVA with line and treatment (saline or formalin) as the independent factors revealed that formalin injection significantly altered *Cnr1* expression in the HA [F(1,38) = 6.7; *p* ≤ 0.05] and the LA mice [F(1,38) = 5.2; *p* ≤ 0.01] (Table 1). An increase in *Cnr1* mRNA was noticed in the amygdala (control: 1.04 ± 0.14; formalin: 2.08 ± 0.18; *p* ≤ 0.001), spinal cord (control: 1.02 ± 0.10; formalin: 1.88 ± 0.29; *p* ≤ 0.001) and hypothalamus (control: 1.0 ± 0.05; formalin: 1.27 ± 0.10; *p* ≤ 0.05) of the HA mice, while the LA line only showed a minor increase in *Cnr1* expression in the hypothalamus (control: 1.02 ± 0.11; formalin: 1.49 ± 0.11; *p* ≤ 0.05).

Differences in Cnr2 expression were also observed following formalin injection in the HA [F(1,35) = 6.5; *p* ≤ 0.05] and LA mice [F(1,38) = 9.2; *p* ≤ 0.001, Two-way ANOVA with line and treatment (saline or formalin) as the independent factors] (Table 2). In the HA mice, an increase in Cnr2 mRNA levels was noted only in the amygdala (control: 1.01 ± 0.08; formalin: 2.08 ± 0.10; *p* ≤ 0.001). In the LA mice, Cnr2 expression increased from 1.07 ± 0.21 to 2.03 ± 0.43 in the periaqueductal gray matter and from 1.02 ± 0.11 to 1.49 ± 0.11 in the hypothalamus.

### 2.5. Opioid Receptor Expression in HA and LA Mice

No changes in opioid receptor expression following formalin injection were evidenced with two-way ANOVA with line and treatment (saline or formalin) as the independent factors in brain structures other than the periaqueductal gray matter. When *Oprm1* expression was studied in the periaqueductal gray matter, two-way ANOVA revealed that formalin injection significantly decreased *Oprm1* mRNA expression in the LA (1.01 ± 0.06 vs. 0.74 ± 0.16; *p* ≤ 0.05), but not in the HA mice [F(1,21) = 9.8; *p* ≤ 0.01] (Table 3).

This finding was confirmed by a significant line x treatment interaction [F(1,21) = 6.6; *p* ≤ 0.05]. As for *Oprd1*, the expression of this gene was higher in the naïve HA than in the LA mice (1.09 ± 0.25 vs. 0.48 ± 0.12; *p* ≤ 0.01) [F(1,11) = 12.4; *p* ≤ 0.01]. Upon formalin injection, *Oprd1* mRNA levels decreased in the periaqueductal gray matter (1.24 ± 0.15 vs. 0.68 ± 0.11; *p* ≤ 0.05) only in the HA mice as revealed by a significant effect of line [F(1,21) = 6.9; *p* ≤ 0.05] and a positive line x treatment interaction [F(1,21) = 6.5; *p* ≤ 0.05] (Table 4).

Of note, a minor increase in *Oprk1* expression (1.19 ± 0.12 vs. 1.48 ± 0.03; *p* ≤ 0.05) was evidenced in the HA line after formalin injection [F(1,21) = 10.0; *p* ≤ 0.01]. This effect was only seen in the HA mice as revealed by a significant line × treatment interaction [F(1,21) = 7.6; *p* ≤ 0.05] (Table 5).

### 2.6. Analysis of 2-AG and AEA in HA and LA Mice

As revealed by one-way ANOVA, formalin injection significantly increased 2-AG and AEA expression in the spinal cord in the HA [2-AG: F(1,12) = 6.85; *p* ≤ 0.05; AEA: F(1,12) = 7.42; *p* < 0.05] but not in the LA mice [2-AG: F(1,12) = 1.35; *p* = 0.27; AEA: F(1,12) = 7.42; *p* = 0.14] (Figure 4). This was confirmed by two-way ANOVA with a significant line x treatment interaction for 2-AG [F(1,24) = 8.20; *p* ≤ 0.01] and AEA levels [F(1,24) = 5.20; *p* ≤ 0.05]. Of note, the HA mice showed lower baseline 2-AG levels compared with their LA counterparts [F(1,12) = 30.0; *p* ≤ 0.001].

### 2.7. Involvement of Microglial Activation in Inflammatory Pain Sensitivity

To assess the role of microglial activation as a potential mechanism responsible for the differences in inflammatory pain sensitivity between both lines, the HA and LA mice were injected with minocycline 30 min prior to formalin injection. As evidenced by two-way ANOVA with phase and treatment as the independent factors, minocycline significantly decreased paw licking time in both the HA [F(2,44) = 12.5; *p* ≤ 0.001] and LA mice [F(2,44) = 9.9; *p* ≤ 0.001] (Figure 5). Only the 100 mg/kg dose of minocycline was effective. In the HA line, minocycline did not have any effect in the interphase, but dose-dependently reduced licking time in the second phase (255 ± 24 s vs. 72 ± 5 s, *p* ≤ 0.001), as evidenced by a positive phase x treatment interaction [F(2,42) = 13.4; *p* ≤ 0.001]. In the LA line, minocycline significantly reduced the intensity of paw licking both in the interphase (95 ± 14 s vs. 34 ± 9 s, *p* ≤ 0.05) and the second phase (90 ± 11.4 s vs. 56.3 ± 8.9 s, *p* ≤ 0.05). Thus, the effect of minocycline was not only phase- but also line-dependent as confirmed by a positive treatment x phase x line interaction [F(2,84) = 8.2; *p* ≤ 0.001].

Two-way ANOVA with line and treatment (saline or formalin) as the independent factors revealed that, in the interphase, formalin affected Aif1 mRNA levels in the amygdala [F(1,15) = 8.26; *p* ≤ 0.01] (Table 6). Namely, relative Aif1 expression increased from 1.49 ± 0.79 to 4.69 ± 0.68 (*p* ≤ 0.05) in the HA mice, while its level was unchanged in the LA line. However, two-way ANOVA failed to detect any significant line x treatment interaction [F(1,15) = 2.70; *p* = 0.12] (Table 6).

Two-way ANOVA with line and treatment (saline or formalin) as the independent factors evidenced no changes in Il-1β expression following formalin injection in any of the examined brain structures (Table 7). However, in the spinal cord, formalin elevated Il-1β mRNA levels in the HA mice as opposed to the LA mice, where Il-1β levels significantly decreased. This bidirectional pattern of Il-1β expression between the HA and LA mice was confirmed by a significant line x treatment interaction [F(1,16) = 8.18; *p* ≤ 0.05] and the main effect of line [F(1,16) = 9.24; *p* ≤ 0.01]. A downward trend for Il-1β expression was visible in the LA mice, but it did not reach significance. The expression of anti-inflammatory M2 microglial marker Il-10 was below the detection level.

## 3. Discussion

In the study herein presented, we explained, at least in part, the possible mechanisms underlying the divergent susceptibility to acute inflammatory pain in mice selected for high (HA) and low (LA) swim-stress-induced analgesia (SSIA). Our results were in accordance with an early study by Lutfy and colleagues [31] who were the first to demonstrate that HA mice were more sensitive to inflammatory pain than LA mice in the formalin test. We observed a similar pattern in the present study, where an intraplantar injection of formalin produced more profound nocifensive behaviours in the HA than the LA mice in the tonic phase (Figure 1). Differential levels of microglial activation could serve as one possible mechanism, as minocycline more profoundly suppressed formalin-induced paw licking in the HA than in the LA mice (Figure 5). This theory was supported by higher mRNA levels of pro-inflammatory markers *Aif1* and *Il-1β* in the HA mice as early as 15 min post-formalin, while their levels were either unchanged or decreased in the LA line (Table 6 and Table 7). The *Aif1* gene encodes the Iba-1 protein that regulates actin cross-linking, which is essential for morphological changes in spinal microglia during the early stages of activation and its expression correlates with hypersensitivity. As *Aif1* expression is also present in ramified microglia, *Il-1β,* a hallmark neuroinflammatory marker, was additionally studied [32,33,34].

We further enhanced the original study by Lutfy [31] by additionally measuring pain thresholds in the interphase of the formalin test. The interphase is characterized by the cessation of excessive neuronal firing provoked by an irritant, such as formalin. Henry et al. argued that this phase is governed by a yet unknown spinally mediated active inhibition mechanism [35]. Our current investigation clearly showed that the HA mice presented significantly less nocifensive behaviours in the interphase than the LA mice (Figure 1). Thus, we hypothesized that this active inhibitory mechanism could be attributed to the endocannabinoid system, as our previous studies indicated substantial differences in its activity between HA and LA mice [17,36]. The HA mice showed a significant increase in spinal and amygdalar *Cnr1* transcripts (Table 1), while *Cnr2* mRNA levels were elevated only in the amygdala (Table 2). Additionally, increased concentrations of anandamide (AEA) and 2-AG were detected in the spinal cord (Figure 4). Moreover, rimonabant—a selective Cnr1 receptor antagonist—dose-dependently intensified pain behaviour in the interphase, but only in the HA line (Figure 3).

Compelling evidence exists for a compensatory peripheral and central increase in AEA and 2-AG in many inflammatory states such as arthritis and osteoarthritis as well as in relapsing multiple sclerosis patients [37,38]. In addition, enhanced AEA signaling resulting from fatty-acid amide hydrolase (FAAH) loss of function was associated with pain insensitivity [3]. These findings imply a significant role of endocannabinoids in neuronal excitability and may account for the divergence in inflammatory pain sensitivity between the HA and LA mice seen in the interphase. Their effect is executed by either a direct action at the Cnr1 receptors and/or modulation of ion channel conductance. It has been recognized that endocannabinoids inhibit Ca^2+^ influx in a Cnr1-dependent and independent manner [39] and also inhibit the tetrodotoxin (TTX)-sensitive Na^+^ channels in cortical and hippocampal neurons [40,41]. Moreover, endocannabinoids activate inwardly rectifying K^+^ channels and induce neuron hyperpolarization [42].

It is commonly acknowledged that spinal Cnr1 receptors are important modulators of nociceptive transmission as their intrathecal delivery produces antinociception in numerous models of pain [43,44,45]. Namely, their activation reduces dorsal horn C- and Aδ-fiber electrical activity both in anesthetized animals and in animals with peripheral inflammation [46,47]. Moreover, loss of spinal Cnr1 receptors produces hyperalgesia [48]. Endocannabinoid binding to Cnr1 receptors in the amygdala is important for the activation of the descending inhibitory pain pathway. Namely, they enhance glutamatergic neurotransmission by suppressing GABA-ergic inhibitory interneurons and increase the firing of periaqueductal gray matter (PAG) projection neurons. This allows stronger input from the rostral ventromedial medulla (RVM) “off” cells to the spinal cord dorsal horn and triggers analgesia. This mechanism is responsible for the effect of WIN 55212-2 microinjection into the amygdalar basolateral nucleus, where it suppresses formalin-induced pain behaviour in the first and tonic phases [49]. Moreover, overexpression of Cnr1 and Cnr2 receptors in the amygdala could possibly serve as an adaptive anxiolytic response to a stressful stimulus such as irritant injection. For instance, it was demonstrated that mice overexpressing Cnr2 receptors were more resistant to stressful stimuli and that Cnr1 receptors mediate the anxiolytic effects of drugs [50,51]. Additionally, stimulation of both Cnr receptor subtypes was implicated in the early, first line mechanism impairing fear memory consolidation and generalization to facilitate aversive memory extinction [52].

To our surprise, nocifensive behaviours seen in the tonic phase in the HA line were not dependent on Cnr1 receptor activation as even the highest dose of rimonabant failed to intensify inflammatory pain in this line. It is possible that increased levels of AEA and 2-AG paired with increased receptor availability in HA mice could alleviate the pronociceptive effect of Cnr1 blockage. Another hypothesis assumes that Cnr1 receptor blockage triggers the compensatory activation of Cnr2 receptor-mediated signaling in the amygdala and in the spinal cord. As shown before, even low-grade inflammation is sufficient to induce anandamide synthesis and enhance spinal Cnr2 receptor expression [53]. The sole role of the Cnr2 receptors in pain sensitivity was evidenced before in nerve injury models, where Cnr2 knockouts showed enhanced ipsilateral pain sensitivity and also developed mirror hyperalgesia [54,55]. Despite no differences in post-formalin spinal Cnr2 expression in HA mice, these mostly microglia-residing receptors, could still be a possible target for spinal 2-AG binding to produce antihyperalgesia in conditions of Cnr1 inhibition, especially when HA mice showed more pronounced microglial activation reflected by an increase in *Aif1* and Il-1β transcripts in response to formalin than LA mice. Moreover, AEA release could be an attempt to reduce microglial polarization towards the inflammatory M1 phenotype, which is Cnr2-receptor-dependent [21]. Additionally, mice from the LA line show impaired G-protein Cnr2 receptor coupled function [17], hence a more prevalent involvement of Cnr1 receptor signaling in the tonic phase and compensatory upregulation of Cnr2 receptors in the PAG and hypothalamus.

The contribution of the endogenous opioid system to inflammatory pain sensitivity in HA and LA mice was also addressed. However, naloxone was ineffective in both lines and both phases (Figure 2), thus providing evidence of the endogenous opioid system not playing a major role in modulation of formalin-induced acute inflammatory pain. One possible explanation for the compromised contribution of the endogenous opioid system in formalin-induced pain sensitivity is the downregulation of *Oprd1* expression in the PAG of HA mice (Table 4). As we have previously shown, δ-opioid receptors significantly influenced nociceptive thresholds in HA mice. For instance, HA mice showed higher basal hypothalamic *Oprd1* expression than their LA counterparts (Table 4). In addition, we have previously identified an A107V substitution in the *Oprd1* gene that renders it less responsive to ligands [56]. An upregulation of the *Oprk1* gene coding the κ-opioid receptor was also evidenced in the PAG of HA mice (Table 5). This phenomenon could be associated with the triggering of aversion circuits in response to a stressful stimulus such as formalin injection [57,58]. HA mice are more vulnerable to stress than LA mice, thus acute stress could induce Oprk1-mediated adaptive changes to trigger aversive and defensive behaviours. The ineffectiveness of naloxone in LA mice may be explained by the widely recognized opioid system hypoactivity and opioid ligand insensitivity in this line [36,59]. Additionally, as shown in the current study, formalin induced *Oprm1* downregulation in the PAG in LA mice, which could also contribute to naloxone insensitivity (Table 3).

## 4. Materials and Methods

### 4.1. Reagents and Kits

Rimonabant was purchased from Tocris Bioscience (Bristol, UK). Naloxone was obtained from Sigma-Aldrich (Taufkirchen, Germany). Formalin (36.5–38% formaldehyde solution in phosphate buffered saline) and chloroform were purchased from POCH S.A (Gliwice, Poland). Zirconium beads (1.4 mm in diameter) for tissue homogenization were purchased from DNA Gdansk (Gdansk, Poland). The RNeasy Lipid Tissue Mini Kit (QIAGEN, Germany) was used to extract RNA from selected brain structures. The Transcriptor First Strand cDNA Synthesis Kit (Roche Diagnostics GmbH, Mannheim, Germany) was used for reverse transcription. Real-time PCR (qPCR) was conducted with the use of TaqMan^®^ Unviersal Master Mix II with UNG and pre-designed TaqMan primers: *Cnr1* (Mm01212171_s1), *Cnr2* (Mm02620087), *Oprm1* (Mm01188089_m1), *Oprd1* (Mm01180757_m1), *Oprk1* (Mm01230885_m1) and *Actb* (Mm02619580_g1) (Applied Biosystems Inc., Waltham, MA, USA). The following reagents for sample preparation for LC-MS/MS were used: phenylmethanesulfonyl fluoride (Sigma Aldrich, Taufkirchen, Germany), toluene and acetone from Avantor Performance Materials Poland (Gliwice, Poland). Internal standards were used: 2-Arachidonoyl Glycerol-d5 (2-AG-d5) and Arachidonoyl Ethanolamide-d4 (AEA-d4) both from Cayman Chemical (Ann Arbor, MI, USA). The reference standards 2-Arachidonoyl Glycerol (2-AG) and Arachidonoyl Ethanolamide (AEA) were bought from TRC, Canada. Solvents for LC-MS (formic acid, acetonitrile HPLC Gradient Grade) were purchased from Merck (Darmstadt, Germany).

### 4.2. Animals

10–12-week-old male Swiss–Webster mice selected for 101 generations for high (HA) and low (LA) swim-stress-induced analgesia were used in behavioural studies. Mice were kept at the Institute of Genetics and Animal Biotechnology Polish Academy of Sciences animal facility under standard ambient temperature (22 ± 2 °C) and humidity (55 ± 10%) under a 12/12 h light/dark cycle (lights on at 7 a.m.). Mice were housed in groups of 3–4 in conventional shoebox cages with sawdust bedding and environmental enrichment. Animals had unlimited access to fresh tap water and pelleted food (Labofeed H, Kcynia, Poland). Experiments were carried out according to the 2010/63/UE directive and received ethical clearance from the I Local Ethics Committee for Animal Experimentation in Warsaw (permit no. WAW2/134/2021). Eight to thirteen animals were used per treatment group. One animal was used only once and a total of 210 animals were used in the study.

### 4.3. Formalin Test

For formalin injection, mice were put under isoflurane anesthesia (Forane, Baxter Deerfield, IL, USA) delivered by an MSS-3 vaporizer (periVet, Szczejkowice, Poland) as described earlier [60]. The flow of isoflurane (in oxygen) was set to 5% (induction) and 3.5% for maintenance. Next, 20 μL of 5% formalin (formaldehyde diluted with phosphate buffered saline) was injected into the dorsum of the right hindpaw with a Hamilton syringe. Due to animal welfare concerns, mice remained under anesthesia for 5 min until the most painful phase of the test (first phase) had resolved. Next, mice were placed in Plexiglas observation chambers mounted on a glass pane surface and the duration of paw licking was manually scored with a stopwatch. Paw licking time was measured in the 5–20 and 20–60 min timeframes representing the interphase and second (tonic) phase respectively in 5-min intervals. The experimenter performing the experiments was blinded to both the line of the animals tested as well as treatment. Twenty-two animals were used to determine formalin sensitivity between HA and LA mice.

### 4.4. Antagonist Administration

Rimonabant (a selective Cnr1 receptor antagonist) was dissolved in a mixture of DMSO/Tween80/saline (1:1:18 *v*/*v*). Naloxone (a nonselective opioid receptor antagonist) was dissolved in saline. Both compounds were delivered intraperitoneally (i.p.) at 100 µL/10 g body weight, 15 min before formalin injection. Control animals received vehicles for rimonabant (a mixture of 5% DMSO, 5% Tween80 and 90% saline) or naloxone (in saline). The solutions were prepared by a laboratory technician and coded to ensure that the experimenter was blind to the treatment. Animals were assigned to different drug treatments by means of a randomization calculator provided by GraphPad QuickCalcs online tool (GraphPad Software, San Diego, CA, USA). A total of 40 animals received naloxone treatment; 72 were treated with rimonabant, 48 with minocycline and 50 received vehicle.

### 4.5. Tissue Harvest

Mice were sacrificed by decapitation under isoflurane (FORANE, Baxter Deerfield, IL, USA) anaesthesia delivered with an MSS-3 vaporizer (periVet, Szczejkowice, Poland). Fifteen minutes following formalin injection, their brains were removed and placed on ice on a Petri dish. Next, the following structures were isolated using the Mouse Brain Matrix (AgnTho’s, Lidingo, Sweden): periaqueductal gray matter, amygdala, hypothalamus and thalamus. The spinal cord was isolated by means of hydraulic extrusion. Briefly, the spinal column was cut at the level of cauda equine and an 18 G needle attached to a 5 mL syringe was inserted into the spinal canal. Hydraulic pressure was applied, and the spinal cord was flushed with ice-cold phosphate-buffered saline onto a clean Petri dish. Next, the lumbar enlargement was located and separated from the sacral and thoracic regions. All central nervous system (CNS) structures were placed on dry ice following dissection and stored at −80 °C. Fourteen animals each were used for qPCR and LC/MS/MS analyses.

### 4.6. RNA Isolation

RNA isolation was carried out with the RNeasy Lipid Tissue Mini Kit (QIAGEN, Germany) according to the manufacturer’s instructions. Tissues were homogenized with the FastPrep-24 homogenizer (MP Biomedicals, Irvine, CA, USA) for 20 s at 4 m/s at room temperature in tubes filled with Zirconium beads (1.4 mm, DNA Gdansk, Poland) and RNA was isolated on mini spin columns according to the enclosed protocol. Quantity and purity of isolated RNA were measured on NanoDrop™ 2000/c (Thermo Fischer Scientific, Waltham, MA, USA).

### 4.7. Reverse Transcription

cDNA synthesis was carried out with the use of the Transcriptor First Strand cDNA Synthesis Kit (Roche Diagnostics GmbH, Mannheim, Germany) following the manufacturer’s protocol. Before cDNA synthesis, the RNA concentration of each sample was adjusted and equalized to the lowest concentration obtained during isolation (45.9 ng/μL). cDNA synthesis (total volume: 20 µL) was carried out with the use of the Transcriptor First Strand cDNA Synthesis Kit (Roche Diagnostics GmbH, Germany) following the manufacturer’s protocol in a PTC-200 thermocycler (MJ Research Inc., Saint-Bruno-de-Montarville, QC, Canada).

### 4.8. Real-Time Polymerase Chain Reaction (qPCR)

Gene expression assessment was carried out with TaqMan^®^ probes (Applied Biosystems Inc., Waltham, MA, USA): *Cnr1* (Mm01212171_s1), *Cnr2* (Mm02620087), *Oprm1* (Mm01188089_m1), *Oprd1* (Mm01180757_m1), *Oprk1* (Mm01230885_m1), *Aif1* (Mm00479862_g1), *Il1β* (Mm00434228_m1), *Il6* (Mm01210733_m1) and *Il10* (Mm01288386_m1). The *Actb* gene (Mm02619580_g1) was chosen as the reference gene based on the NormFinder software. The reaction was carried out in a mixture containing 5 μL TaqMan^®^ Unviersal Master Mix II with UNG, 3.5 μL RNase- and DNase-free water, 0.5 μL pre-designed TaqMan probes and 1 μL of cDNA template. The reaction was performed in a LightCycler^®^ 96 Real-Time PCR System (Roche Diagnostics GmbH, Mannheim, Germany) under the following conditions: UNG incubation (50 °C, 120 s) and polymerase activation (95 °C, 600 s) followed by 45 cycles of denaturation (95 °C, 15 s) and hybridization/elongation (60 °C, 60 s). To measure the accumulation of the fluorescent signal, threshold cycle values (Ct) were determined for each sample. The comparative method (CNRQ = 2^−ΔΔCt^), represented as x-fold expression, was used to determine mRNA levels of the analysed genes [61].

### 4.9. LC-MS/MS Analysis

The tissue samples (15–70 mg) were weighed and kept on ice. For every 1 mg of tissue, 10 μL of ice-cold acetone with 1 mM phenylmethanesulfonyl fluoride and 20 μL of internal standards solution (50 ng/mL of 2-AG-D5 and 5 ng/mL of AEA-D4) was placed. Then, the samples were homogenized on ice using a probe sonicator. Time of sonication was adjusted to the tissue type. The homogenate was centrifuged (10,000× *g*, 5 min, 4 °C). Then, the supernatant was extracted with toluene (1:1, *v*/*v*) using vortex (4 min, 1800 rpm) and a waterbath sonicator (0.5 min). After the centrifugation (10,000× *g*, 2 min, 4 °C), the organic layer was evaporated under a stream of nitrogen (20 °C, 8 min) and the dry residue was reconstituted with 70 μL of acetonitrile and injected into the LC-MS/MS.

Instrumental analysis was performed using an Agilent 1260 Infinity system (Agilent Technologies, Santa Clara, CA, USA) coupled to QTRAP 4000 (AB Sciex, Framingham, MA, USA). The turbo ion spray source was operated in the positive mode. The curtain gas, ion source gas 1, ion source gas 2 and collision gas (all high purity nitrogen) were set at 241 kPa, 414 kPa, 276 kPa and “high” instrument units (4.6 × 10^−5^ Torr), respectively. The ion spray voltage and source temperature were 5500 V and 600 °C, respectively. The target compounds were analysed in multiple reaction monitoring (MRM) mode. The quantitative MRM transitions, declustering potential (DP) and collision energy (CE) for 2-AG, 2-AG-D5, AEA and AEA-D4 were (*m*/*z*) 379 > 287 (DP = 91 V, CE = 21 V), 384 > 287 (DP = 91 V, CE = 21 V), 348 > 62 (DP = 81 V, CE = 39 V) and 352 > 66 (DP = 81 V, CE = 33 V), respectively. Chromatographic separation was achieved with a Kinetex^®^ C18 column (100 mm × 4.6 mm, 2.6 µm) from Phenomenex (Torrance, CA, USA). The column was maintained at 20 °C at a flow rate of 0.5 mL min-1. The mobile phases consisted of 0.2% formic acid as eluent A and acetonitrile with 0.2% formic acid as eluent B. The gradient (%B) was as follows: 0 min 75%, 0.5 min 75%, 1.5 min 90% and 5 min 90%. The re-equilibration of the column to the initial conditions lasted 1.8 min. The injection volume was 10 μL. The analytical procedure allowed separation of 1-arachidonoylglycerol (1-AG) and 2-arachidonoylglycerol (2-AG) and prevented 1-AG/2-AG isomerization during sample preparation [62].

### 4.10. Statistical Analysis

Data were processed with STATISTICA 13.3 software (TIBCO Software Inc., Palo Alto, CA, USA). Graphs were created with GraphPad Prism 5.04 software for Windows (GraphPad Software, San Diego, CA, USA). Behavioural and receptor expression data were analysed with two-way ANOVA with phase and treatment or line and phase as independent factors, followed by Bonferroni’s post hoc-test and expressed as means ± SEM. One-way ANOVA was used for comparisons within each line for each phase separately. LC-MS/MS analysis of 2-AG and anandamide concentrations in the brain structures of HA and LA mice were processed with one-way ANOVA with Bonferroni’s post hoc-test and expressed as means ± SEM. The statistician was blinded to the line of the animal data and treatment. Normality was assessed with the Shapiro–Wilk test.

## 5. Conclusions

In conclusion, this was the first study that underlines the possible mechanisms underpinning the differences in inflammatory pain sensitivity between HA and LA mice. The possible mechanism behind decreased pain behaviour in HA mice observed in the interphase encompasses enhanced Cnr1-dependent inhibitory signaling related to both enhanced spinal 2-AG and AEA release along with increased *Cnr1* gene transcription. Conversely, the pronociceptive response of HA mice in the tonic phase was not modulated by Cnr1 receptor activation and was mediated by increased microglial activation. Endogenous opioid system activation exerted no evident modulatory function regarding inflammatory pain sensitivity.

## Figures and Tables

**Figure 1 ijms-23-11686-f001:**
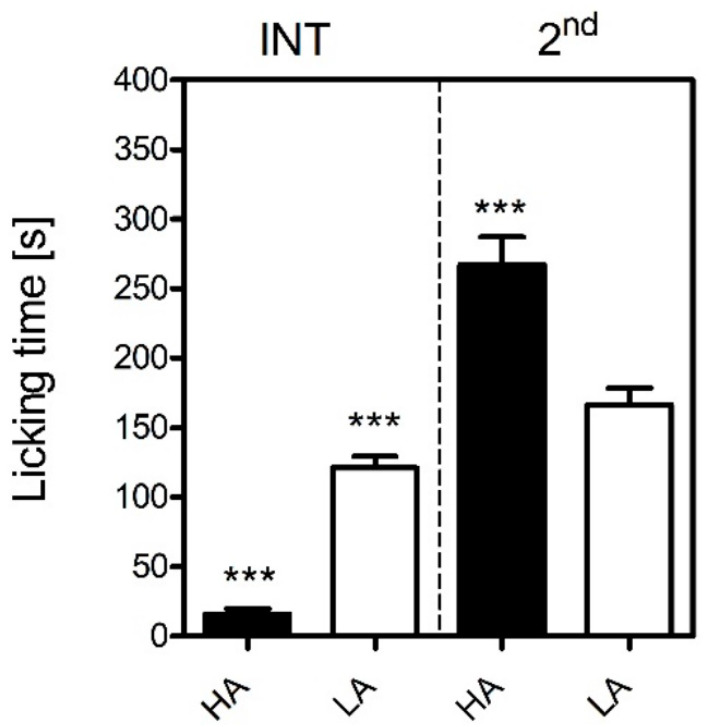
Intensity of acute inflammatory pain in HA and LA mice (*n* = 9–13) in the formalin test upon injection of 5% formalin (in PBS) into the dorsum of the hind foot. Paw licking time was measured in the interphase (INT, 5–20 min) and the second phase (2nd, 20–60 min) of the test. Results were analysed with two-way and one-way ANOVA, followed by Bonferroni’s post-hoc test for multiple comparisons. Statistical significance was depicted as follows: *** *p* ≤ 0.001 (HA vs. LA within each phase).

**Figure 2 ijms-23-11686-f002:**
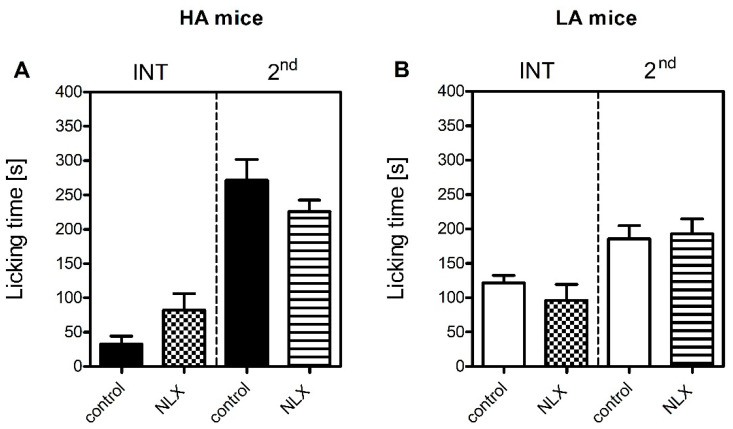
The effect of naloxone (1 mg/kg, i.p., NLX) on acute inflammatory pain sensitivity in HA (**A**) and LA (**B**) mice (*n* = 8–10) in the formalin test upon injection of 5% formalin (in PBS) into the dorsum of the hind foot. Paw licking time was measured in the interphase (INT, 5–20 min) and the second phase (2nd, 20–60 min) of the test. NLX was administered 15 min prior to formalin. Results were analysed within each line and phase with two-way and one-way ANOVA followed by Bonferroni’s post-hoc test. No statistical significance was observed.

**Figure 3 ijms-23-11686-f003:**
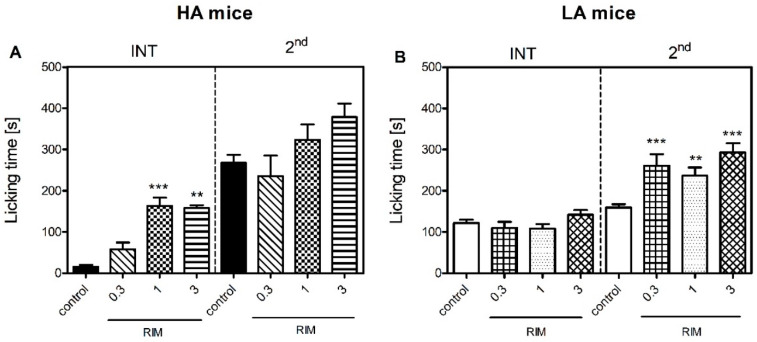
The effect of rimonabant (0.3–3 mg/kg, i.p., RIM) on acute inflammatory pain sensitivity in HA (**A**) and LA (**B**) mice in the formalin test (*n* = 9–13) upon injection of 5% formalin (in PBS) into the dorsum of the hind foot. RIM was administered 15 min prior to formalin. Paw licking time was measured in the interphase (INT, 5–20 min) and the second phase (2nd, 20–60 min) of the test. Results were analysed with two-way ANOVA with treatment and phase as independent factors and with one-way ANOVA within each phase for each mouse line, followed by Bonferroni’s post-hoc test. Statistical significance was depicted as follows: ** *p* ≤ 0.01; *** *p* ≤ 0.001 (vs. control within each phase and line).

**Figure 4 ijms-23-11686-f004:**
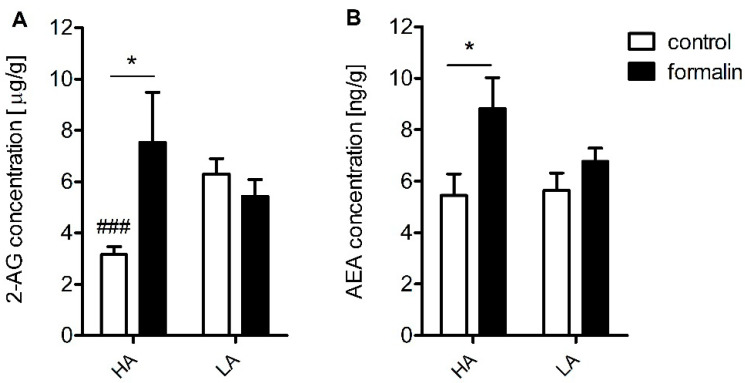
LC-MS/MS analysis of spinal cord 2-arachidonoylglycerol (2-AG) (**A**) and anandamide (AEA) concentration (**B**) in HA and LA mice (*n* = 7) injected with either formalin (5% in PBS) or PBS. The effect of formalin injection within each line was analysed with one-way ANOVA followed by Bonferroni’s post-hoc test. Between-line interaction was assessed with two-way ANOVA and Bonferroni’s post-hoc test. Statistical significance was depicted as follows: * *p* ≤ 0.05 (vs. control within the line); ### *p* ≤ 0.001 (HA vs. LA).

**Figure 5 ijms-23-11686-f005:**
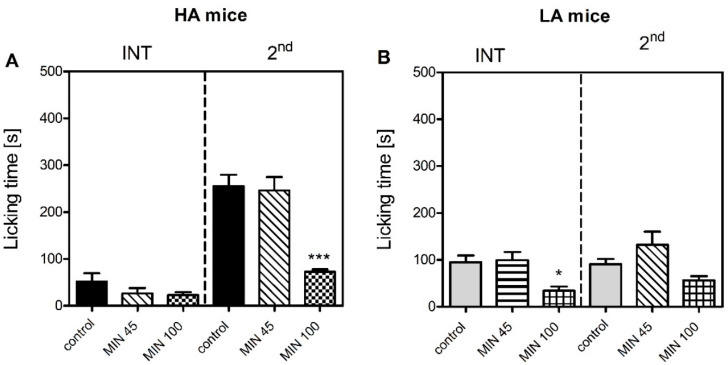
Effect of minocycline (45 and 100 mg/kg, i.p, MIN) on the intensity of acute inflammatory pain in HA (**A**) and LA (**B**) mice (*n* = 8) in the formalin test. Paw licking was scored following intraplantar injection of 5% formalin (in PBS) into the dorsum of the hind foot. Minocycline was injected 30 min before formalin. Paw licking time was measured in the interphase (INT, 5–20 min) and the second phase (2nd, 20–60 min) of the test. Results were analysed within each line and phase with two-way ANOVA followed by Bonferroni’s post-hoc test. Statistical significance was depicted as follows: * *p* ≤ 0.05; *** *p* ≤ 0.001 (vs. control within each phase).

**Table 1 ijms-23-11686-t001:** Cannabinoid receptor 1 (Cnr1) mRNA levels in HA and LA mice in the interphase of the formalin test. Mice were injected with 5% formalin (in PBS) into the dorsal aspect of the hind paw (*n* = 4–6). Samples were harvested 15 min post-formalin. Results were analysed within each line with two-way ANOVA followed by Bonferroni’s post-hoc test. Statistical significance was depicted as follows: * *p* ≤ 0.05; *** *p* ≤ 0.001 (vs. control).

Structure	HA Control	HA Formalin	LA Control	LA Formalin
periaqueductal grey matter	1.01 ± 0.06	1.43 ± 0.34	1.01 ± 0.09	1.94 ± 0.64
amygdala	1.04 ± 0.14	2.08 ± 0.18 ***	1.02 ± 0.10	0.85 ± 0.17
spinal cord	1.02 ± 0.1	1.88 ± 0.29 ***	1.06 ± 0.19	1.25 ± 0.31
hypothalamus	1.00 ± 0.05	1.27 ± 0.10 *	1.02 ± 0.11	1.49 ± 0.11 *
thalamus	1.02 ± 0.11	0.94 ± 0.15	1.00 ± 0.06	1.28 ± 0.24

**Table 2 ijms-23-11686-t002:** Cannabinoid receptor 2 (Cnr2) mRNA levels in HA and LA mice in the interphase of the formalin test. Mice were injected with 5% formalin (in PBS) into the dorsal aspect of the hind paw (*n* = 4–6). Samples were harvested 15 min post-formalin. Results were analysed within each line with two-way ANOVA followed by Bonferroni’s post-hoc test. Statistical significance was depicted as follows: * *p* ≤ 0.05; *** *p* ≤ 0.001 (vs. control).

Structure	HA Control	HA Formalin	LA Control	LA Formalin
periaqueductal grey matter	1.21 ± 0.39	2.08 ± 0.54	1.07 ± 0.21	2.03 ± 0.43 *
amygdala	1.01 ± 0.08	2.08 ± 0.32 ***	0.90 ± 0.13	1.10 ± 0.24
spinal cord	1.37 ± 0.20	1.36 ± 0.49	1.01 ± 0.08	1.21 ± 0.18
hypothalamus	1.03 ± 0.14	1.34 ± 0.39	0.80 ± 0.12	1.34 ± 0.15 *
thalamus	1.20 ± 0.40	1.50 ± 0.34	1.04 ± 0.16	1.08 ± 0.19

**Table 3 ijms-23-11686-t003:** Mu opioid receptor 1 (Oprm1) mRNA levels in HA and LA mice in the interphase of the formalin test. Mice were injected with 5% formalin (in PBS) into the dorsal aspect of the hind paw (*n* = 4–6). Samples were harvested 15 min post-formalin. Results were analysed within each line with two-way ANOVA followed by Bonferroni’s post-hoc test. Statistical significance was depicted as follows: * *p* ≤ 0.05 (vs. control).

Structure	HA Control	HA Formalin	LA Control	LA Formalin
periaqueductal grey matter	1.1 ± 0.12	1.16 ± 0.10	1.01 ± 0.06	0.74 ± 0.16 *
amygdala	0.99 ±0.07	1.65 ± 0.25	1.16 ± 0.15	0.91 ± 0.27
spinal cord	1.02 ± 0.12	1.37 ± 0.27	1.01 ± 0.06	0.73 ± 0.12
hypothalamus	1.00 ± 0.04	1.01 ± 0.07	1.01 ± 0.22	1.08 ± 0.21
thalamus	1.01 ± 0.08	1.02 ± 0.12	1.02 ± 0.10	1.12 ± 0.11

**Table 4 ijms-23-11686-t004:** Delta opioid receptor 1 (Oprd1) mRNA levels in HA and LA mice in the interphase of the formalin test. Mice were injected with 5% formalin (in PBS) into the dorsal aspect of the hind paw (*n* = 4–6). Samples were harvested 15 min post-formalin. Results were analysed within each line with two-way ANOVA followed by Bonferroni’s post-hoc test. Statistical significance was depicted as follows: * *p* ≤ 0.05 (vs. control); ## *p* ≤ 0.01 (vs. control LA).

Structure	HA Control	HA Formalin	LA Control	LA Formalin
periaqueductal grey matter	1.24 ± 0.15	0.68 ± 0.11 *	1.15 ± 0.25	2.18 ± 0.82
amygdala	0.6 ± 0.06	0.82 ± 0.15	1.19 ± 0.40	1.14 ± 0.15
spinal cord	1.09 ± 0.28	0.53 ± 0.15	1.15 ± 0.25	0.83 ± 0.42
hypothalamus	1.09 ± 0.25 ##	1.15 ± 0.28	0.48 ± 0.12	1.06 ± 0.21
thalamus	1.06 ± 0.20	1.02 ± 0.12	1.09 ± 0.21	1.1 ± 0.12

**Table 5 ijms-23-11686-t005:** Kappa opioid receptor 1 (Oprk1) mRNA levels in HA and LA mice in the interphase of the formalin test. Mice were injected with 5% formalin (in PBS) into the dorsal aspect of the hind paw (*n* = 4–6). Samples were harvested 15 min post-formalin. Results were analysed within each line with two-way ANOVA followed by Bonferroni’s post-hoc test. Statistical significance was depicted as follows: * *p* ≤ 0.05 (vs. control).

Structure	HA Control	HA Formalin	LA Control	LA Formalin
periaqueductal grey matter	1.19 ± 0.12	1.48 ± 0.03 *	1.04 ± 0.13	0.82 ± 0.20
amygdala	1.78 ± 0.31	1.84 ± 0.23	1.15 ± 0.17	0.74 ± 0.10
spinal cord	1.07 ± 0.26	1.89 ± 0.58	1.04 ± 0.13	0.76 ± 0.13
hypothalamus	1.02 ± 0.14	1.04 ± 0.14	0.77 ± 0.15	1.08 ± 0.19
thalamus	1.01 ± 0.05	1.14 ± 0.38	1.01 ± 0.07	1.11 ± 0.09

**Table 6 ijms-23-11686-t006:** Allograft inflammatory factor 1 (Aif1) mRNA levels in HA and LA mice in the interphase of the formalin test. Mice were injected with 5% formalin (in PBS) into the dorsal aspect of the hind paw (*n* = 4–6). Samples were harvested 15 min post-formalin. Results were analysed within each line with two-way ANOVA followed by Bonferroni’s post-hoc test. Statistical significance was depicted as follows: * *p* ≤ 0.05 (vs. control).

Structure	HA Control	HA Formalin	LA Control	LA Formalin
periaqueductal grey matter	1.76 ± 0.99	0.68 ± 0.19	1.08 ± 0.56	1.09 ± 0.19
amygdala	1.49 ± 0.79	4.69 ± 0.68 *	1.65 ± 0.12	2.52 ± 0.16
spinal cord	1.00 ± 0.13	1.00 ± 0.14	1.05 ± 0.17	1.03 ± 0.49
hypothalamus	1.02 ± 0.20	1.34 ± 0.14	1.25 ± 0.26	1.93 ± 0.29
thalamus	1.04 ± 0.19	1.63 ± 0.72	1.14 ± 0.16	1.09 ± 0.41

**Table 7 ijms-23-11686-t007:** Interleukin 1β (Il-1β) mRNA levels in HA and LA mice in the interphase of the formalin test. Mice were injected with 5% formalin (in PBS) into the dorsal aspect of the hind paw (*n* = 4–6). Samples were harvested 15 min post-formalin. Results were analysed within each line with two-way ANOVA followed by Bonferroni’s post-hoc test. Statistical significance was depicted as follows: * *p* ≤ 0.05 (vs. control).

Structure	HA Control	HA Formalin	LA Control	LA Formalin
periaqueductal grey matter	1.60 ± 0.86	1.77 ± 0.26	1.17 ± 0.27	1.19 ± 0.25
amygdala	2.63 ± 2.01	0.72 ± 0.12	1.09 ± 0.23	0.76 ± 0.22
spinal cord	1.39 ± 0.40	4.90 ± 1.41 *	1.25 ± 0.44	0.37 ± 0.06 *
hypothalamus	1.36 ± 0.53	3.95 ± 1.30	2.67 ± 1.89	1.59 ± 0.91
thalamus	1.09 ± 0.21	3.87 ± 1.56	1.13 ± 0.28	0.61 ± 0.21

## Data Availability

Not applicable.

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
