# Peer review of "Mechanisms Underlining Inflammatory Pain Sensitivity in Mice Selected for High and Low Stress-Induced Analgesia—The Role of Endocannabinoids and Microglia"

_ijms, 2022, doi:10.3390/ijms231911686_

Round 1
Reviewer 1 Report
In this manuscript Poznanski et al. study involvement of opioid, cannabinoid and microglial responses in the nocifensive responses to a formalin treatment. In my view, the work has important flaws that unfortunately make difficult the intepretation of the results despite the effort put by the authors.
First, there is a lack of a control group of mice that does not show high or low stress induced analgesia. In the absence of this control group, it can’t be assessed whether the observed phenotypes (behavior or gene/cannabinoid expression) are actually altered or not in HA or LA mice. The inclusion of this control group is needed for adequate interpretation of the results.
Another important caveat of the study is that the formalin test, a model of inflammatory/chemically-induced pain extensively used in pain research, is conducted under isoflurane anesthesia and nocifensive responses are recorded immediately afterwards. Then, it is unclear whether the studied treatments may affect the level of consciousness of the animals rather than nociception itself, or whether the effects/duration of isoflurane anesthesia may be different in HA/LA mice or how these isoflurane could affect the nocifensive responses in these animals. I understand the good will of the researchers avoiding the first phase of the formalin test, but the conclusions extracted can be confusing and impede the comparison of the results to previous results obtained in the formalin test. Also, the first phase (0-5 min) is normally associated to direct stimulation of nociceptors and the second phase (around 10-60 min) of the canonical formalin test is related with inflammatory response, and both are separated by an interphase period of absence of activity. The “interphase” shown by the authors involves relatively high levels of activity and it is unclear why they separate between 6-20 and 21-60. A representation of the time-courses may be helpful to see whether there is such a separation.
Minocycline has been used as a glial inhibitor but it is also expected to alter gut microbiome, which has shown also behavioral effects. Assuming that the effects of the antibiotic are due to changes in microglial function based solely on two inflammatory markers seems rather insufficient evidence.
I would suggest also different organization of the manuscript , with behavioral and expression data in the same figure for each of the different cannabinoid/opioid/glial mechanisms studied. This would facilitate overall interpretation of the data.
Reviewer 2 Report
The manuscript studies the possible mechanisms causing the difference of pain perception between high analgesia line and low analgesia line mice using the formalin test. The behavior differences are related to the cannabinoid system in the interphase and microglial activation in the tonic phase. The latter results were underlined with a solid increase of pro-inflammatory markers in the HA line after formalin. The paper is straightforward and the results are very interesting.
Minor issues:
Abstract:
“Mice were intraplantarly injected with 5% formalin” – Mice received intraplantar injections of 5% formalin..
Introduction:
The last sentences of the introduction are already referring to the results/conclusion. Although I enjoyed to read it like this, it is not necessary the accepted academic style.
Results:
The indication of the bars in the graphs is not consistent with the animal lines/tested phase. Please consider the same color with the same animal lines/phases (eg. results of HA animals in the first phase are always white).
Discussion:
I prefer enhanced instead of “enriched” in line 281.
Reviewer 3 Report
The paper by Piotr Poznanski et al. aims to determine whether endocannabinoid system activity could 16 account for the differences in acute inflammatory pain sensitivity in mouse lines selected for high 17 (HA) and low (LA) swim stress-induced analgesia (SSIA). Although my opinion is overall positive I have some comments to make.
Line 38-49: the authors stated that the enhancement of endocannabinoid tone produces an anti-inflammatory phenotype in animal models or pain insensitivity in human subjects but then, after this sentence, only CNR1 and CNR2 antagonists/inverse agonists are cited. Moreover they asserted that the inhibition of FAAH and MAGL leads to endocannabinoid breakdown. I think that this paragraph is very confusing. There are several papers in which pain was studied in vivo with agonists (DOI: 10.1016/j.ejmech.2020.112858) or other modulators of cannabinoid receptors (DOI: 10.1021/acs.jmedchem.8b00368; DOI: 10.1021/acs.jmedchem.2c00582; DOI: 10.1038/npp.2015.148) which are not antagonists or inverse agonists.
Reviewer 4 Report
- This article contains only 45 references. The author’s might add more citations to present the role of opiod and endocannbinoid systems in the pathomachanism of inflammation.
-
- Should be interesting to investigate the effect of kynurenine system in this setting. It is well-known that kynurenines can influence CB1Rs (Zador et al., Long-term systemic administration of kynurenic acid brain region specifically elevates the abundance of functional CB 1 receptors in rats.,2020).
Reviewer 5 Report
This article investigates the involvement of endogenous cannabinoids and opioids in formalin-induced pain in mice selected for high and low stress-induced analgesia.
Criticism
1. Abstract, l.18. It should be indicated already here that the selection of these lines was not a subject of the present paper. It should be mentioned later in the paper, whether the selection took place in the authors Institute or elsewhere. It would be also nice to learn, which was the threshold above which a mouse was considered LA.
2. L.85. Transient neuron hyperpolarization of which neurons? In the case of opioids probably the Nucleus Locus coeruleus.
3. Fig. 1. It should be mentioned already here (it is described in the Methods), why the first phase of formalin-induced pain was not taken into consideration.
4. Statistics. It should be indicated among which values the statistical significance was calculated. This criticism holds true throughout the paper.
5. In Figs. 2, 3, 5 you should indicate in the Fig. that the left panel shows the HA, and the right panel the LA animals. The definition in A, and B is not sufficient.
6. Tables. The contents of all Tables would be more convincing if you showed the data in Figs. There is also no reason to fragment the content of the papers by showing a number of separate panels. Why you do not prepare summary Figs. bringing together these data.
7. L.245. Explain the importance of Aif1 as a marker or cause of inflammation.
8. All qPCR and LC/MS-MS data were made from tissue collected after formalin injection. Thus there is no separation according to first phase-intermediary-second phase responses which are partly of opposite direction.
9. You used only rimodabant as a CB1 antagonist but failed to use a CB2 or mixed CB1-CB2 antagonist.
